# A seven-sex species recognizes self and non-self mating-type via a novel protein complex

**Guanxiong Yan[1], Yang Ma[1], Yanfang Wang[2], Jing Zhang[1], Haoming Cheng[1,3], Fanjie Tan[2], Su Wang[1,3], Delin Zhang[2], Jie Xiong[1,4], Ping Yin[2]\*, Wei Miao[1,3,4,5]\***

[1]Institute of Hydrobiology, Chinese Academy of Sciences, Wuhan, China; [2]National Key Laboratory of Crop Genetic Improvement, Hubei Hongshan Laboratory, Huazhong Agricultural University, Wuhan, China; [3]University of Chinese Academy of Sciences, Beijing, China; [4]Key Laboratory of Breeding Biotechnology and Sustainable Aquaculture, Chinese Academy of Sciences, Wuhan, China; [5]Hubei Hongshan Laboratory, Wuhan, China

**\*For correspondence:**
yinping@mail.hzau.edu.cn (PY);
miaowei@ihb.ac.cn (WM)

**Competing interest:** The authors declare that no competing interests exist.

**Abstract** Although most species have two sexes, multisexual (or multi-mating type) species are also widespread. However, it is unclear how mating-type recognition is achieved at the molecular level in multisexual species. The unicellular ciliate *Tetrahymena thermophila* has seven mating types, which are determined by the MTA and MTB proteins. In this study, we found that both proteins are essential for cells to send or receive complete mating-type information, and transmission of the mating-type signal requires both proteins to be expressed in the same cell. We found that MTA and MTB form a mating-type recognition complex that localizes to the plasma membrane, but not to the cilia. Stimulation experiments showed that the mating-type-specific regions of MTA and MTB mediate both self- and non-self-recognition, indicating that *T. thermophila* uses a dual approach to achieve mating-type recognition. Our results suggest that MTA and MTB form an elaborate multifunctional protein complex that can identify cells of both self and non-self mating types in order to inhibit or activate mating, respectively.

## eLife assessment

This **fundamental** study provides insight into the fascinating process of self- and non-self-recognition in the protist *Tetrahymena thermophila*, a species with seven distinct mating types. Using an elegant combination of phenotypic assays, protein studies, and imaging, the authors present **convincing** evidence that a large multifunctional protein complex at the cell surface mediates both self- and non-self mating-type recognition. This study extends our understanding of how more than two mating types/sexes may be specified in a species, and it will be relevant for anyone interested in sexual systems and cell-cell communication.

## Introduction

Sexual reproduction is almost universal among eukaryotic organisms. Mating type (or sex) is a key regulatory feature of gamete fusion. Most species have only two sexes/mating types (e.g. male and female, + and -, or a and α) and species usually use either self- or non-self-recognition mechanism to achieve sex/mating-type recognition (***Goodenough and Heitman, 2014***). However, species in some lineages, such as some ciliates and basidiomycetes (***Heitman, 2015***; ***Phadke and Zufall, 2009***), possess multiple mating types, and multiple-alleles self-incompatibility system was observed in some

**Figure 1.** Mating-type recognition in *T. thermophila*. (**A**) Example of self and non-self mating-type recognition. When one cell of mating type I encounters another, costimulation and mating do not occur. When a cell of mating type I encounters a cell of another mating type (II–VII), the cells enter the costimulation stage and go on to form a pair. (**B**) Two typical phenotypes of the costimulation stage are Tip transformation and concanavalin A (Con-A) receptor appearance. Yellow dashed circle, transformed cell tip (center, single cell) or pairing junction (right, cell pair). Note that Tip transformation may become less obvious after Con-A staining. (**C**) *MTA* and *MTB* gene structure and MTA and MTB protein domain information (*Cervantes et al., 2013*). MTA and MTB form a head-to-head gene pair. For each gene, the terminal exon is shared by all mating types and the remainder is mating-type-specific (the sequence differs for each mating type). The mating-type-specific region of each protein is predicted to be extracellular.

The online version of this article includes the following figure supplement(s) for figure 1:

**Figure supplement 1.** *T. thermophila* life cycle.

plants, such as Brassicaceae (*Iwano and Takayama, 2012*; *Takayama and Isogai, 2005*; *Vekemans and Castric, 2021*). This raises the interesting question of how sexes/mating types are recognized at the molecular level in multiple sex/mating-type systems.

The model unicellular ciliate, *Tetrahymena thermophila*, has seven mating types (I–VII). Under starvation conditions, any cell of one mating type can mate with a cell of any of the other six mating types, but not with one of the same mating type (*Figure 1A*, *Figure 1—figure supplement 1*, *Videos 1–3*; *Cervantes et al., 2013*; *Nanney, 1953*; *Orias et al., 2017*; *Yan et al., 2021*). Mating-type recognition in *Tetrahymena* depends on direct cell–cell contact (temporary or persistent), which suggests that mating-type proteins localize to the cell surface. However, there is no direct evidence to indicate whether they are ciliary proteins or not. When one cell comes into contact with a cell of a different mating type, a mating-type-dependent recognition event enables both cells to enter a pre-conjugation stage (called costimulation) (*Bruns and Palestine, 1975*; *Finley and Bruns, 1980*). Even when cells of different mating types are mixed in unequal ratios (e.g. 9:1), all cells become fully stimulated (*Bruns and Palestine, 1975*). This is because one cell can temporarily contact a number of cells

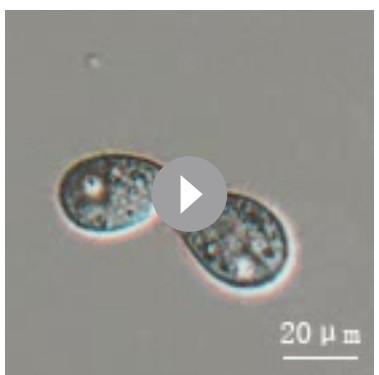

**Video 1.** Mating behavior of *T. thermophila*.
https://elifesciences.org/articles/93770/figures#video1

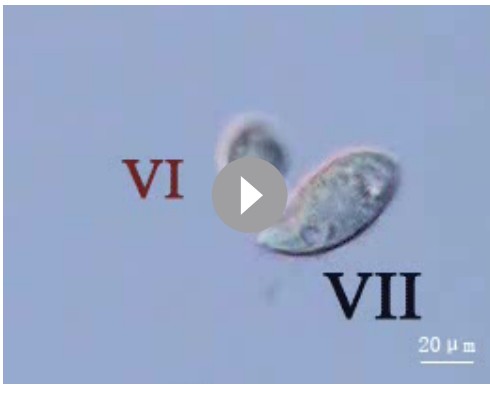

**Video 2.** Mating behavior of *T. thermophila*. To distinguish cells of different mating types, smaller mating type VI cells and larger mating-type VII cells were used in this experiment.
https://elifesciences.org/articles/93770/figures#video2

and stimulate them. Processes that take place during costimulation include Tip transformation (*Wolfe and Grimes, 1979*) and concanavalin A (Con-A) receptor appearance (*Figure 1B*; *Wolfe and Feng, 1988*; *Wolfe et al., 1986*). In preparation for pairing, costimulated cells of the same and different mating type(s) adhere to form very loose pairs. Heterotypic cell pairs form a stable conjugation junction, whereas homotypic pairs separate very quickly (*Videos 2 and 3*; *Kitamura et al., 1986*).

The mating-type system of *T. thermophila* was described by Nanney and collaborators in the early 1950s (reviewed in *Orias, 1981*; *Orias et al., 2017*). We previously showed that mating type is determined by a pair of mating-type genes that are organized in head-to-head orientation: *MTA* and *MTB* (*Figure 1C*; *Cervantes et al., 2013*). Each gene has a terminal exon that encodes five transmembrane (TM) helices and a cysteine-rich growth factor receptor (GFR) domain. The region between the two terminal exons of the gene pair encodes the N-terminal mating-type-specific extracellular regions. Based on the mating-type-specific regions, the mating-type genes are called *MTA1-MTB1* for mating type I, *MTA2-MTB2* for mating type II, and so on. We previously showed that Δ*MTB* cells do not form pairs or produce progeny and that Δ*MTA* cells retain mating-type specificity but pair extremely poorly and rarely produced progeny (*Cervantes et al., 2013*); we concluded that the two genes are nonredundant and both are essential for mating. In addition, our previous work demonstrated that CDK19, CYC9, and CIP1 are coexpressed with MTA and MTB and are essential for mating. These encoded proteins constitute components of a cyclin-dependent kinase complex, which localizes to the cell tip and pairing junction (*Ma et al., 2020*). However, challenges such as multiple mating types; the high molecular weight, membrane localization and extremely low expression levels of mating-type proteins; and difficulty in genetically manipulating the mating-type gene locus have so far prevented elucidation of the mode of action of the MTA and MTB proteins and of whether they mediate self- or non-self mating-type recognition.

In this study, we provide direct evidence that the MTA and MTB form an elaborate multifunctional protein complex that can identify cells of both self and non-self mating types to inhibit or activate mating, respectively.

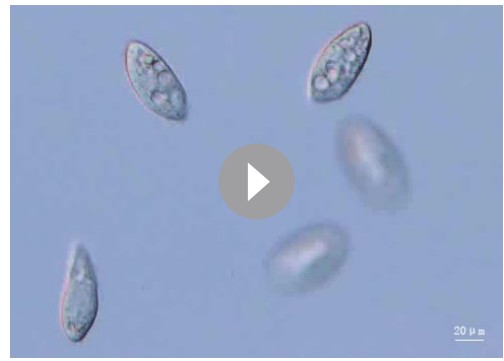

**Video 3.** Cells of different mating types form a pair, whereas cells of the same mating type become separated after a short contact. To distinguish cells of different mating types, smaller mating type VI cells and larger mating-type VII cells were used in this experiment.
https://elifesciences.org/articles/93770/figures#video3

## Results

### Mating-type recognition cannot be explained by the simple receptor–ligand model

Receptor–ligand interaction is a critical mechanism for intercellular communication that may regulate mating-type recognition in *T. thermophila*, irrespective of whether self- or non-self-recognition mechanisms are employed. Therefore, we first assessed whether mating-type recognition conforms to a straightforward receptor–ligand model (in which one individual mating-type protein acts as the receptor and the other as its ligand) and whether self or non-self is recognized (*Figure 2—figure supplement 1A, 1 and 2*). For this, we determined whether deletion of each mating-type gene affected the transmission and detection of mating signals to and from wild-type (WT) cells (*Figure 2A* shows the experimental procedure) by assessing the ability of cells to undergo costimulation (the prerequisite for mating). Our experiment allows us to test if cells missing one of the two mating-type proteins can still costimulate WT cells (for details, refer to *Figure 2—figure supplement 1A, 3–6*).

In *T. thermophila*, cells normally enter into the fully costimulation stage within ~30 min after mixing starved WT cells of two different mating types, and start pair formation during the next ~30 min (*Figure 2B*, black line). Cells that have already been costimulated immediately start forming pairs with other costimulated cells of a different mating type (*Figure 2B*, red line).

To our surprise, the rate of pair formation in WT cells pre-incubated with either ΔMTA cells (*Figure 2B*, green line) or ΔMTB cells (*Figure 2B*, blue line) did not increase (i.e. costimulation did not occur). It indicated that neither the MTA protein from ΔMTB cells nor the MTB protein from ΔMTA cells can stimulate the WT cell, which does not fit any deductions based on the simple receptor–ligand model we proposed (*Figure 2—figure supplement 1A*). These results were not changed by extending the pre-incubation time (*Figure 2—figure supplement 1C*). Therefore, both MTA and MTB proteins are essential for the mating-type signal; there is no simple receptor–ligand relationship.

In addition, WT cells were not costimulated even when they were simultaneously incubated with both ΔMTA and ΔMTB cells (*Figure 2B*, teal line), although, according to the receptor–ligand model, they should have received 'MTA stimulation' from ΔMTB cells and 'MTB stimulation' from ΔMTA cells. This result indicates that the absence of a mating-type protein in one cell cannot be complemented by its presence in another cell in the same culture; the MTA and MTB proteins must be in the same cell to transmit the mating-type signal. This finding also suggests that mating-type recognition cannot be explained by a simple receptor–ligand model. It is possible that the MTA and MTB proteins form a complex which either serves as a recognizer (functioning as both ligand and receptor) or a co-receptor. But, since MTA and MTB are the only genes with mating-type specificity, it is unlikely that the complex is acting as a co-receptor. Whether MTA and MTB act as a ligand and a receptor independently within the complex will be discussed later.

### Mating-type proteins differentially regulate two steps of costimulation

During costimulation, cells undergo a sequence of developmental events that remodel the anterior cell membrane and its associated cytoskeleton (*Cole, 2013*). Two hallmarks of this process are Tip transformation (in which the anterior tip of the cell becomes curved) and Con-A receptor appearance (receptors bound by the plant lectin Con-A, which binds to mannose containing glycoproteins). When WT cells of one mating type were mixed with WT cells of another mating type, the cell tips became transformed (*Figure 2C, 4*) and Con-A receptors appeared in almost all cells (*Figure 2D, 4*). When WT cells were pre-incubated with ΔMTA, ΔMTB, or both cell types, Tip transformation was not observed in any cell (*Figure 2C, 5–7*). Similarly, when cells of each mutant were pre-incubated with WT cells, Tip transformation was not detected (*Figure 2C, 8 and 9*). The outcome was slightly different for Con-A receptor appearance. Con-A receptors were not observed in WT cells pre-incubated with ΔMTB cells (*Figure 2D, 6*) or in cells of either mutant pre-incubated with WT cells (*Figure 2D, 7 and 8*). In contrast, when WT cells were exposed to ΔMTA or 'ΔMTA cells plus ΔMTB cells', the Con-A receptor was detected (*Figure 2D, 5 and 9*); this is consistent with ΔMTA cells retaining a very weak ability to pair (*Cervantes et al., 2013*). These results indicate that neither ΔMTA and ΔMTB cells can fully stimulate WT cells or be stimulated by WT cells. They also demonstrate that costimulation can be separated into two stages: (i) one represented by the appearance of Con-A receptors that only requires MTB protein in partner cells, and (ii) the other represented by morphological transformation of the cell tip, which requires both MTA and MTB.

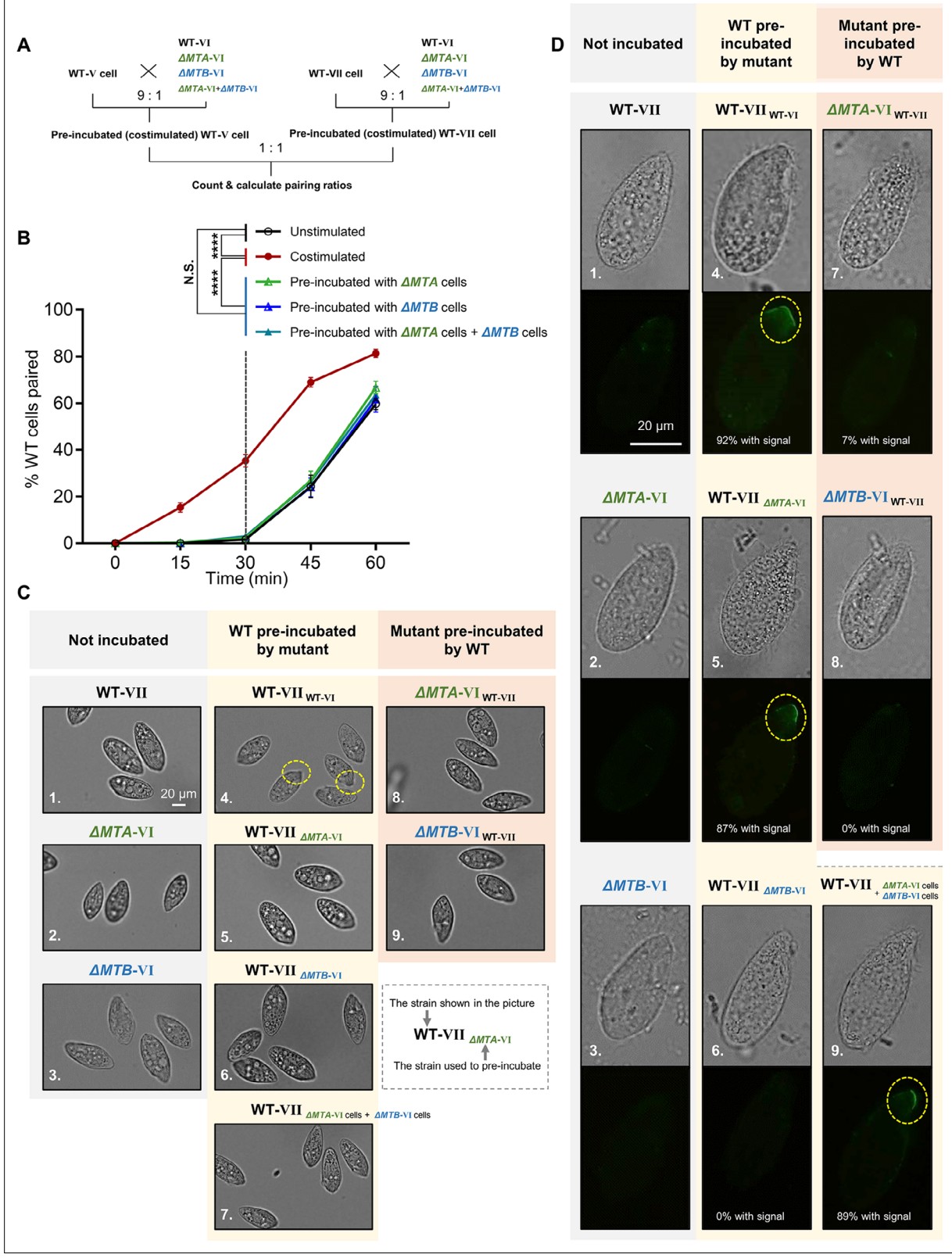

**Figure 2.** Mating-type proteins are essential for mating-type recognition. (**A**) Experimental procedure for the costimulation experiments. Starved wild-type (WT) cells of mating types V (WT-V) and VII (WT-VII) were separately pre-incubated with the indicated mating type VI mutant (9:1 ratio) for 30 min and then the pre-incubated cells were mixed at a 1:1 ratio. Note that before mixing the costimulated cells, any potentially pairing cells were separated by shaking. (**B**) Effect of pre-incubation with ΔMTA and ΔMTB on the rate of pair formation. Each experiment was repeated three times,

*Figure 2 continued on next page*

*Figure 2 continued*

with>100 pairs counted at each time point. Matched two-way ANOVA was used for the statistical analysis. N.S., not significant; *, p<0.05; **, p<0.01; ***, p<0.001; ****, p<0.0001. Unpaired mutants were excluded when calculating the pairing rate (see Materials and methods). (C) Tip transformation, a hallmark of costimulation. Each strain was pre-incubated with the strain shown in subscript. Yellow dashed circle, transformed cell tip. (D) Appearance of concanavalin A (Con-A) receptors, another hallmark of costimulation. In all, ~90% cells show Con-A receptor fluorescence (panels 4, 5, and 9). The low percentage of cells (7%) with fluorescence in panel 7 were probably WT cells, which comprised 10% of the pre-incubation culture. Each strain was pre-incubated with the strain shown in subscript. Yellow dashed circle, Con-A receptor fluorescence.

The online version of this article includes the following figure supplement(s) for figure 2:

**Figure supplement 1.** Mating-type gene deletion strains do not costimulate wild-type (WT) cells.

## Mating-type proteins form a complex with several coexpressed proteins

According to the pre-incubation results with 'ΔMTA cells plus ΔMTB cells', MTA and MTB cannot functionally complement each other when expressed on different cells (*Figure 2B*, teal; *Figure 2C, 7*). In contrast, we previously found that ΔCDK19 and ΔCYC9 cells, which express both MTA and MTB, cannot mate. Pre-incubating WT cells with each of them of different mating types promotes WT cells mating (*Ma et al., 2020*). The difference in results between these two types of pre-incubation may be caused by whether MTA and MTB proteins are expressed on the same or different cells: Since TM helices fix the proteins onto the cytomembrane, the mating-type proteins are limited on each cell. Thus, in ΔMTA and ΔMTB cells, the remaining mating-type protein expressed on different cells (MTB or MTA individually) is likely to be spatially separated. WT cells, even when pre-incubated with both ΔMTA and ΔMTB cells simultaneously, receive MTB or MTA signals separately (*Figure 2—figure supplement 1G*). However, in ΔCDK19 and ΔCYC9 cells, MTA and MTB proteins are expressed on the same cell, allowing them to provide MTA and MTB signals together (*Figure 2—figure supplement 1E*). These findings lead us to propose the hypothesis that MTA and MTB proteins form a mating-type recognition complex (MTRC).

To test whether MTA and MTB proteins physically interact, an HA-tag coding sequence was ligated to the 3′ end of the *MTA* gene (*Figure 3A*); cellular proteins were co-precipitated with HA-tagged MTA and analyzed by immunoprecipitation-coupled mass spectrometry (IP-MS). As expected, the MTB protein co-purified with MTA (*Figure 3B and C*), as did another set of proteins, which we named MRC1–MRC6 (*Figure 3B and C* and *Figure 3—source data 1 and 2*). Next, we produced strains expressing either HA-tagged MTB or MRC1 (*Figure 3A*), and found that each protein pulled down a subset of the proteins that co-purified with MTA (*Figure 3B and C*). Unfortunately, these pull-down experiments were not as successful as the MTA IPs, perhaps because of the higher molecular weight of MTB (194 kDa) and MRC1 (212 kDa). Taken together, our results suggest that MTA, MTB, and MRC1–MRC6 form the MTRC. Alternatively, MTA and MTB may interact with subsets of MRC proteins to form smaller complexes or alternative MTRCs. Different protein interactors were identified in extracts from cells at different mating stages. This may reflect conformational changes in the MTRC but the huge molecular weight of the complex and extremely low expression levels of its proteins make this possibility difficult to investigate.

All the identified components of MTRC are large membrane proteins (predicted size, 92–212 kDa). *Figure 3D* shows the predicted domains of the MRC1–MRC6 proteins. Like MTA and MTB, MRC1 has five predicted TM helices and a GFR domain. MRC2 has eight TM helices and a pectin lyase-fold domain, suggesting a possible role in carbohydrate chain modification. MRC3 has two TM helices in the central region and an adjacent ~35 amino acid (aa) poly-E region. MRC4 and MRC5 (previously named TPA9; *Wang et al., 1997*; *Wang and Takeyasu, 1997*) are both P-type ATPases that are likely to function as calcium-translocators. MRC6 has four TM helices and a P-loop containing nucleoside triphosphate hydrolases. Most of the MRC genes are highly coexpressed with *MTA* and *MTB* (*Figure 3E*). Examination of the whole genome sequences of strains with mating types II–VII confirmed that only the sequences of *MTA* and *MTB* genes are mating-type-specific.

In addition to the MRC proteins, CDK19, CYC9, CIP1, and AKM3 were identified in IP-MS experiments, but with relatively few supporting peptides (*Figure 3C*). CDK19, CYC9, and CIP1 have been proved to be essential for mating (*Ma et al., 2020*). AKM3 is predicted to be a K⁺ channel of unknown biological function. It is coexpressed with *MTA* and *MTB* (*Figure 3E*), and we found that the *AKM3*

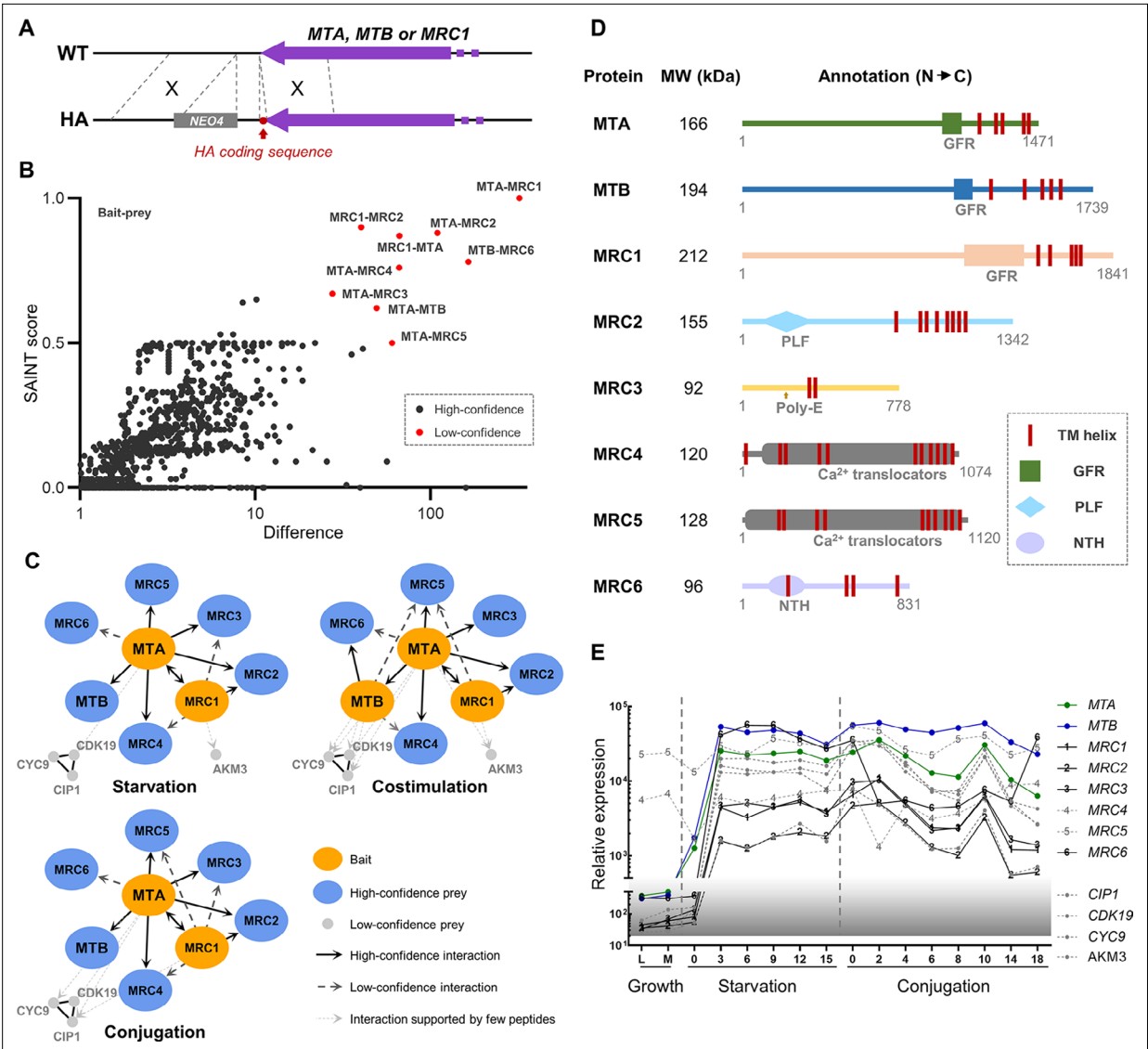

**Figure 3.** Proteins that interact with MTA and MTB. (**A**) Construction of HA-tagged strains. All of the tagged strains mated like wild-type (WT) cells. (**B**) Statistical analysis of immunoprecipitation-coupled mass spectrometry (IP-MS) data. A total of 13 experiments were carried out. WT samples (untagged) were run in parallel for each sample. All 13 WT controls were combined as the background control. Red dot, high-confidence interaction; dark gray dot, low-confidence interaction. Gene identifiers are summarized in *Figure 3—source data 2*. Note that the wash buffer contained 1% Triton X-100 and 600 mM NaCl. (**C**) Interaction network based on IP-MS data. Orange oval, bait; blue oval, high-confidence prey; light gray dot, low-confidence prey; black line, high-confidence interaction; dark gray dashed line, low-confidence interaction; light gray dotted line, interaction supported by a few peptides (these proteins were shown because their coding genes are coexpressed with *MTA* and *MTB* and deleting them affects mating behavior). (**D**) Diagram of functional domain annotation of mating-type recognition complex (MTRC) components. GFR, growth factor receptor domain; PLF, pectin lyase fold; Poly-E, poly-glutamic acid region; NTH, P-loop-containing nucleotide triphosphate hydrolase. (**E**) Expression profiles of genes whose protein products were identified by IP-MS as potentially components of the MTRC. Expression data is derived from *Tetra*FGD (*Xiong et al., 2011*).

The online version of this article includes the following source data for figure 3:

**Source data 1.** Immunoprecipitation-coupled mass spectrometry (IP-MS) results.

**Source data 2.** Gene identifiers.

deletion strain cannot pair. Therefore, these four proteins are also likely to interact with the mating-type proteins (perhaps indirectly and/or via weak interactions) and might be involved in downstream signaling following mating-type recognition.

## Mating-type proteins localize to the cell surface but not to the cilia

We used the MTA7-HA strain to determine the localization of mating-type proteins. Cell fractionation (*Figure 4—figure supplement 1*) revealed that MTA7-HA is a membrane protein (*Figure 4A*). Biotinylation and isolation of cell surface proteins confirmed that MTA7-HA is exposed on the cell surface (*Figure 4B*). To investigate whether MTA7-HA localizes to the cilia membrane, we isolated and collected cilia (*Figure 4C*) and then analyzed cilia protein extracts by IP-coupled western blotting (WB) and MS. MS analysis identified typical ciliary proteins, such as inner and outer arm dynein proteins (*Figure 4—source data 1*). However, both IP-WB (*Figure 4D*) and MS (*Figure 4—source data 1*) consistently failed to identify MTA7-HA protein in the cilia fraction. These results conclusively indicate that MTA7-HA localizes to the cell surface, but not to cilia. Unfortunately, we failed to detect MTA7-HA by immunofluorescence staining of cells at any mating stage (starvation, costimulation, or conjugation), probably due to the epitope masking and extremely low expression level.

To further examine localization of the mating-type proteins, eGFP-tagged MTB2 was overexpressed from an exogenous locus (*Figure 4E*). This strain (which has a mating type VI background) mated normally with WT cells of all mating types except for VI and II. This result indicates that the overexpressed MTB2-eGFP protein is fully functional for mating. Interestingly, cells of this strain can also mate with one another (self mating); a similar phenotype was previously reported for strains expressing multiple mating-type proteins (*Lin and Yao, 2020*). These selfing ability may be caused by the interaction between heterotypic MTRCs. The overexpressed MTB2-eGFP protein was detected on the cell surface in a linear pattern radiating from the cell tip to the cell body along the ciliary rows (*Figure 4F*, costimulated cell; *Figure 4G*, mating pair), although signals are also apparent between ciliary rows. Co-staining with a tubulin dye showed that the MTB2-eGFP protein is adjacent to, rather than co-localizing with, the base of cilia (*Figure 4F and G*, enlarged). Confocal images from the interior of the cell and through the cilia showed that MTB2-eGFP localizes to the cell surface (and also to intracellular structures, probably the endoplasmic reticulum [ER] and Golgi), but not to the cilia (*Figure 4—figure supplement 2*), confirming our results with the MTA7-HA protein (*Figure 4A–D*). MS analysis showed that overexpressed MTB2-eGFP protein was not present in isolated cilia of the *MTB2-eGFP* strain (*Figure 4—source data 2*). Therefore, mating-type proteins localize to the cell surface, as might be expected since mating-type recognition depends on cell–cell contact.

## Mating-type proteins influence non-self-recognition

The mating-type-specific region of the *MTA* and *MTB* gene pair is the only known genetic locus with mating-type specificity; therefore, we next tested whether this region influences self- and/or non-self mating-type recognition. For this, we expressed the extracellular region of MTA or MTB protein (MTAxc or MTBxc, respectively) in an insect cell secreted expression system, purified the recombinant proteins (*Figure 5—figure supplement 1*), and then tested their effect on mating behavior.

First, we tested whether MTAxc and MTBxc can influence mating in cells with a different mating-type specificity (i.e. non-self-recognition). When WT cells were incubated with MTAxc (and/or MTBxc) of a different mating type, markers of costimulation were not observed (*Figure 5—figure supplement 2*). Surprisingly, treated cells had a significantly increased pairing rate. Compared with controls (*Figure 5A–C*, black), WT cells (VI and VII) pre-treated with MTAxc or MTBxc of different mating types (VII and VI, respectively) had a similar increased pairing rate (*Figure 5*, green or blue). It with a stronger increasing effect after pre-treatment with both MTAxc and MTBxc (that is, MTA and MTB have synergistic effect) (*Figure 5A–C*, teal), consistent with our finding that MTA and MTB interact with each other (*Figure 2—figure supplement 1F*). Dose–effect assays showed that pairing rates increased with increasing MTA6xc and MTB6xc concentrations between 3 pg/ml and 30 pg/ml, with the effect becoming saturated or weaker at higher concentrations (*Figure 6A–F*). MTAxc and MTBxc also stimulated mating for all other WT mating types (*Figure 6G and H*), indicating that this is a general effect. Based on these results, we conclude that the mating-type-specific regions of MTA and MTB proteins are involved in non-self mating-type recognition.

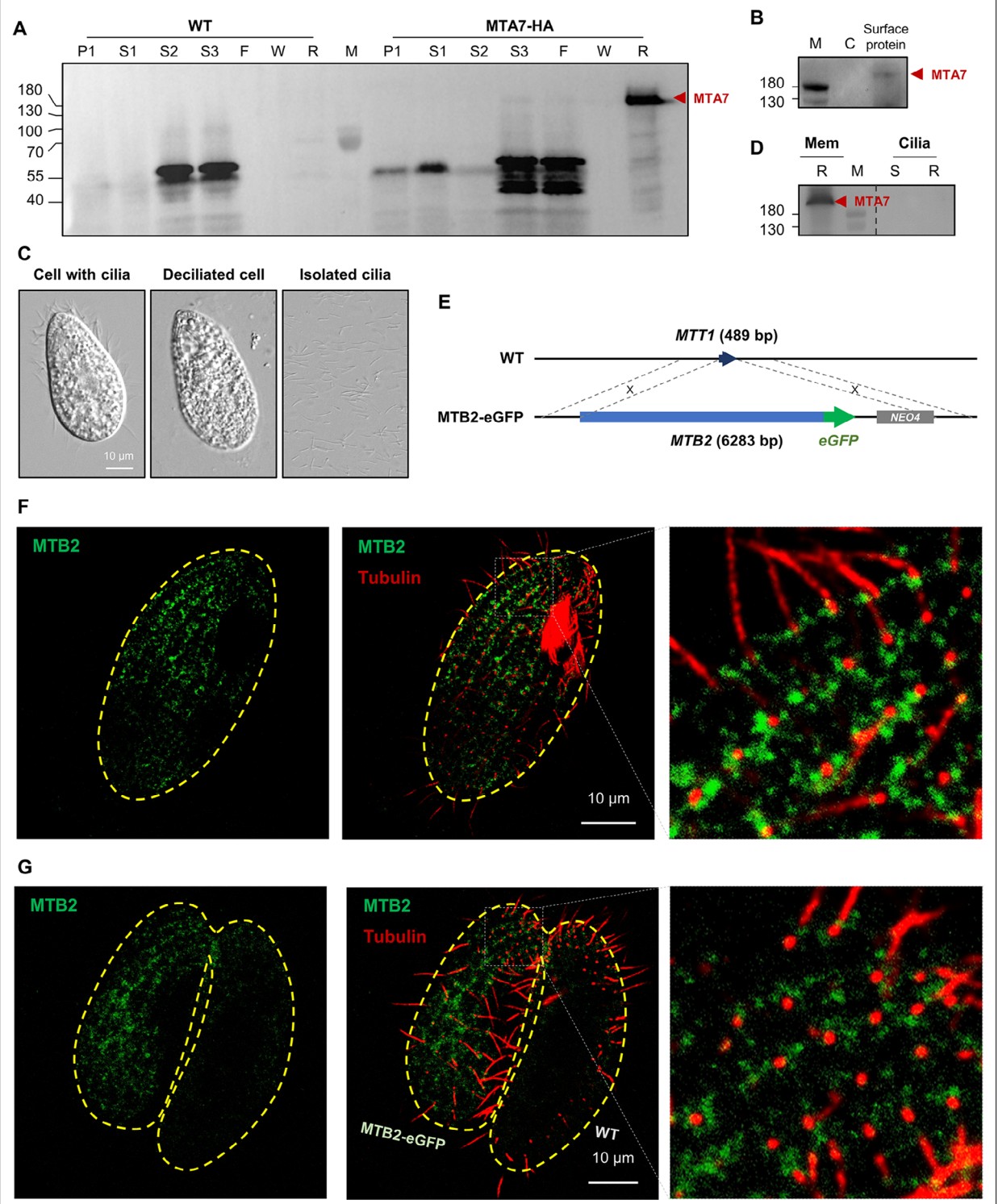

**Figure 4.** Mating-type proteins are cell surface proteins but do not localize to cilia. (**A**) Fractionation of *MTA7-HA* cells (please see *Figure 4—figure supplement 1A* for the experimental process). Red arrowhead, MTA7-HA; F, flow through; P, pellet; R, resin; S, supernatant; W, wash. The MTA signal is undetectable until S3 (enriched membrane proteins), and only appears after affinity chromatography (R). (**B**) Western blotting (WB) analysis of cell surface proteins. Red arrowhead, MTA7-HA; M, marker; C, negative control (unbiotinylated). (**C**) Cilia isolation and purification. (**D**) WB analysis of IP products of membrane and ciliary proteins. Mem, membrane; R, resin; S, supernatant. The same amount of *MTA7-HA* cells was used for the membrane and ciliary protein IPs. The full blot is shown in *Figure 4—figure supplement 1B*. (**E**) Construction scheme for eGFP-tagged MTB2 strains.

*Figure 4 continued on next page*

*Figure 4 continued*

(**F**) Costimulated *MTB2-eGFP* cell. (**G**) Paired *MTB2-eGFP* × WT cell. To induce MTB2-eGFP overexpression, cells were treated with 10 ng/ml Cd$^{2+}$ for 5 hr. Green, eGFP signal; red, tubulin signal; yellow dashed line, cell outline. The focal plane of these images is the cell surface.

The online version of this article includes the following source data and figure supplement(s) for figure 4:

**Source data 1.** MS analysis of MTA7-HA cilia protein.

**Source data 2.** MS analysis of *MTB2-eGFP* cilia protein.

**Source data 3.** TIF containing *Figure 4A* and original scan of the relevant western blot analysis (anti-HA) with highlighted bands and sample labels.

**Source data 4.** TIF containing *Figure 4B* and original scan of the relevant western blot analysis (anti-HA) with highlighted bands and sample labels.

**Source data 5.** TIF containing *Figure 4D*, *Figure 4—figure supplement 1B*, and original scan of the relevant western blot analysis (anti-HA) with highlighted bands and sample labels.

**Figure supplement 1.** Fractionation of MTA7-HA cells.

**Figure supplement 2.** Confocal images of ciliary sections and cell interior sections of *MTB2-eGFP* cells.

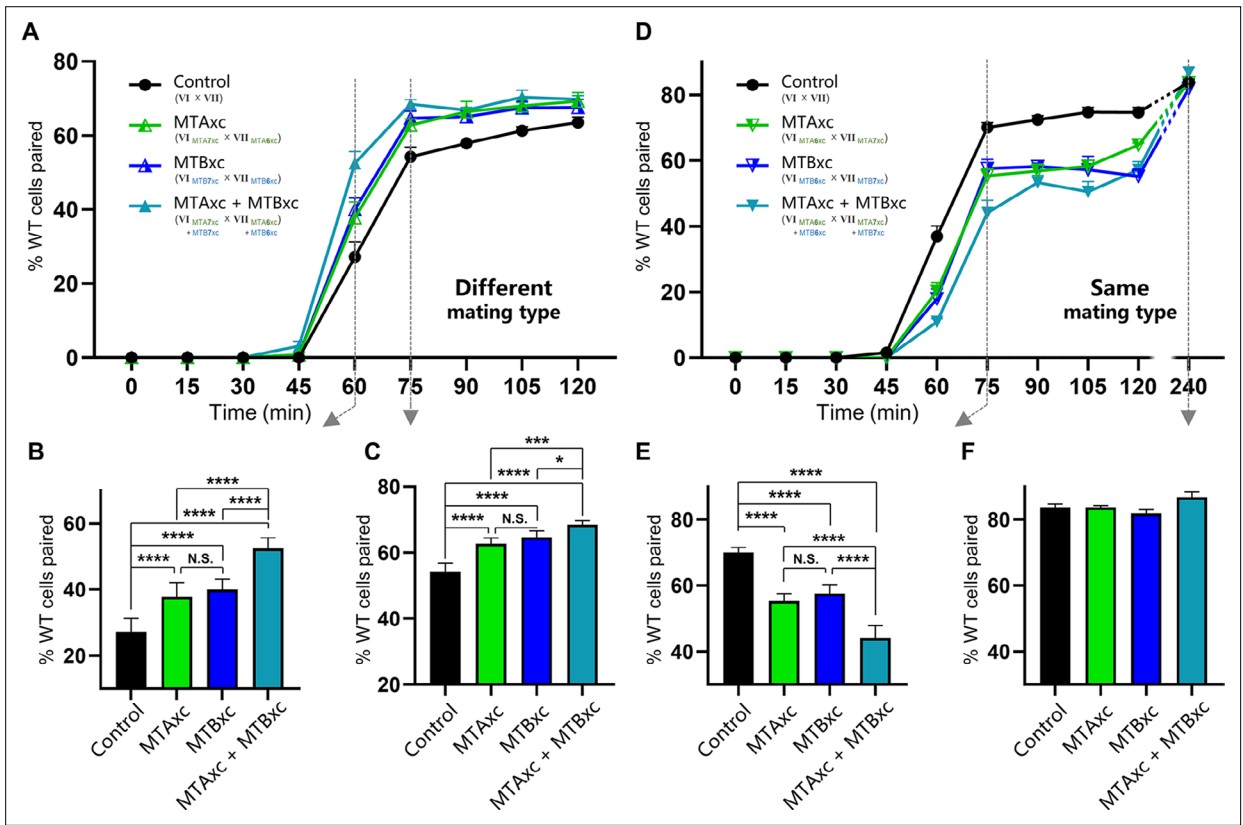

**Figure 5.** Stimulation experiments using MTAxc and/or MTBxc. (**A**) Wild-type (WT) cells were treated with MTAxc and/or MTBxc proteins (30 pg/ml, 1 hr) of different mating-type specificities. WT-VI cells were treated with MTA/B7xc protein, and WT-VII cells were treated with MTA/B6xc protein. Treated cells were washed twice before mixing to remove residual proteins from the starvation medium. Note that the starvation medium used for washing should contain mating-essential factors secreted by *T. thermophila* cells during starvation (*Adair et al., 1978*). The mating types used in each experiment is shown in the figure. Each strain was pre-incubated with the strain shown in subscript. Each experiment was repeated five times, with >100 pairs counted at each time point. Matched two-way ANOVA was used for the statistical analysis. N.S., not significant; *, p<0.05; **, p<0.01; ***, p<0.001; ****, p<0.0001. Error bars, SEM. (**B, C**) The percentages of cells paired at 60 min (**B**) and 75 min (**C**) were used for the statistical analysis (method described in *Figure 2B*). (**D**) WT cells (mating types VI and VII) were treated with MTAxc and/or MTBxc proteins of the same mating-type specificity (as described in A). (**E, F**) The percentages of cells paired at 75 min (**E**) and 240 min (**F**) were used for the statistical analysis (method described in *Figure 2B*).

The online version of this article includes the following figure supplement(s) for figure 5:

**Figure supplement 1.** Expression and purification of extracellular regions of mating-type proteins.

**Figure supplement 2.** Treatment with MTA6xc and/or MTB6xc proteins fails to induce costimulation.

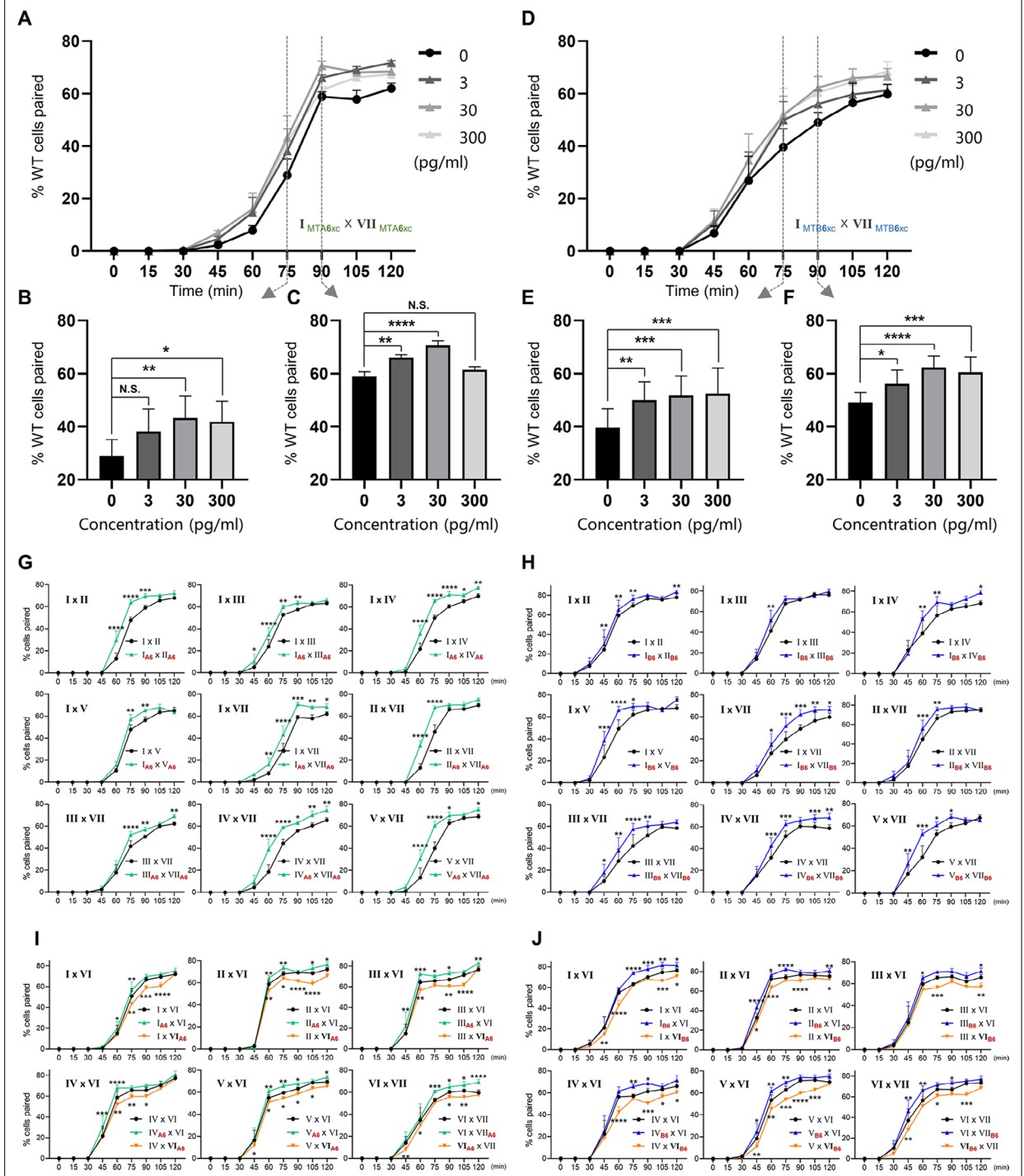

**Figure 6.** Results of treatment with either MTAxc or MTBxc proteins. (**A–F**) Dose–response effect of treatment with MTA6xc or MTB6xc protein. (**A–C**) MTA6xc results. (**D–F**) MTB6xc results. Cells of mating types I and VII were used for these experiments. Experimental and statistical methods were as described for *Figure 5*, except for protein concentrations. (**G–J**) MTA6xc or MTB6xc proteins affect the mating of various combinations of other WT mating types. (**G**) MTA6xc results. Both mating partners were treated. (**H**) MTB6xc results. Both mating partners were treated. (**I**) MTA6xc results. Cells of only one mating type were treated. Note that mating type VI cells were used in these experiments. (**J**) MTB6xc results. Note that mating type VI cells were used in these experiments. Each experiment was repeated five times, with >100 pairs counted at each time point. Matched two-way ANOVA was used for the statistical analysis. N.S., not significant; *, p<0.05; **, p<0.01; ***, p<0.001; ****, p<0.0001. Error bars, SEM. Experimental and statistical methods were as described for *Figure 5*. The mating types used in each experiment are shown in the figure. Red subscript 'A6' or 'B6' in (**G–J**) indicates the strain treated with MTA6xc or MTB6xc, respectively.

These results also shed light on whether MTA and MTB act independently as a ligand and a receptor within the complex. For instance, if MTA is a ligand and MTB is a receptor, treating a cell with MTAxc protein should induce a mating signal, whereas treatment with MTBxc should not. However, our results indicate that MTAxc and MTBxc have very similar effects, and the effect is stronger when treated with two proteins together. Therefore, this possibility is unlikely. In addition, we did not identify any costimulation markers when incubated with MTAxc and/or MTBxc. This may be due to the lack of other components, differences in post-translational modifications between insect cells and *Tetrahymena*, or variations in protein conditions between the cell membrane and solution.

## Mating-type proteins also influence self-recognition

We used similar methods to examine whether pre-treatment with a cognate mating-type-specific region (i.e. self-recognition) affects mating. Treatment of WT cells (VI and VII) with MTAxc and/or MTBxc of the same mating type decreased the pairing rate (*Figure 5D and E*, green or blue). No obvious difference was found between treatments with MTAxc and MTBxc. Meanwhile, a significant synergistic effect was observed (*Figure 5D and E*, teal). For all treatments (single or combined), the pairing rate was similar by 4 hr (reaching >80%; *Figure 5D and F*), indicating that the initial inhibitory effect on pairing was eventually overcome. Negative regulation by MTAxc and MTBxc was also observed for other mating-type combinations (*Figure 6I, J*). These results support the idea that the

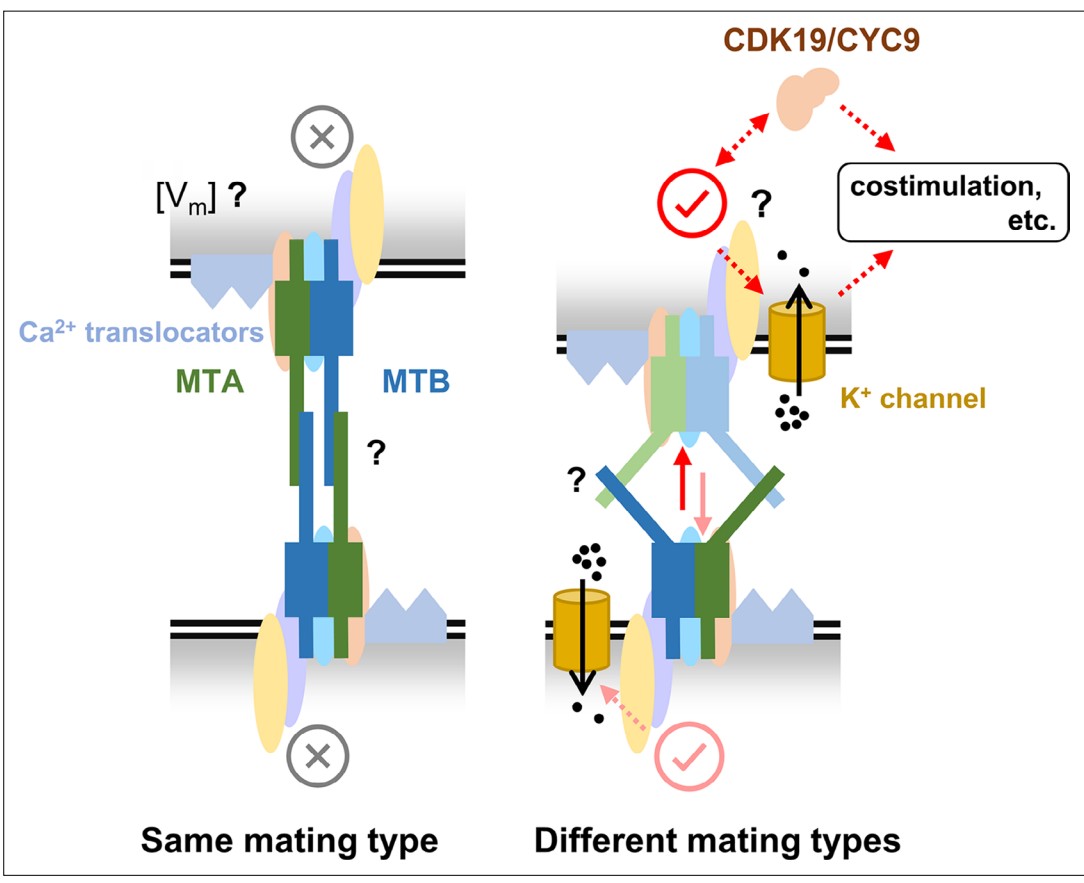

**Figure 7.** A hypothesized mating-type recognition model. MTA and MTB function by forming mating-type recognition complex (MTRC) with several other proteins. When cells of the same mating type contact each other, the interaction between MTRCs inhibits mating. Conversely, when cells of different mating types contact each other, the interaction between MTRCs initiates mating. Many details remain unknown, such as (i) during self-recognition, whether the MTRC is only blocked or generates an inhibitory signal (e.g. membrane potential by $Ca^{2+}$), (ii) how two MTRCs interact during cell–cell recognition, and the differences between self- and non-self-recognition, and (iii) the downstream pathway when MTRC is activated. We hope future studies will help refine and advance this model, contributing to a comprehensive understanding of how mating types are recognized in multiple mating systems.

mating-type-specific regions of MTA and MTB proteins mediate both non-self (between different mating types) and self (between the same mating type) recognition.

## Discussion

Although the basic biological features of the *T. thermophila* mating-type system were discovered over half a century ago, the mechanism for mating-type recognition remains unclear. Here, we identified a novel MTRC that contains MTA, MTB, and several other proteins and provide evidence that MTA and MTB mediate both self and non-self mating-type recognition (*Figure 7*).

An open question is why *T. thermophila* should use such a dual approach (both self and non-self) to achieve mating-type recognition. Recent research on basidiomycetes and flowering plants (other species with multiple mating types) has shown that their mating-type recognition (or self-incompatibility) mechanisms involve either self- or non-self-recognition (*Fraser and Heitman, 2003*; *Iwano and Takayama, 2012*; *Vekemans and Castric, 2021*). Our previous evolutionary study showed that the length of the mating-type-specific region differs significantly among different *Tetrahymena* species (~2000 aa for *T. thermophila*, ~3200 aa for *T. borealis*, while only ~1000 aa for *T. pigmentosa*) (*Yan et al., 2021*). Based on the massive difference in length, it is reasonable to speculate that different species might use different mechanisms for mating-type recognition. Therefore, dramatic evolution of the mating-type recognition mechanism seems to have occurred relatively soon after the emergence of the *Tetrahymena* genus. Further detailed functional and evolutionary studies may reveal whether the mating-type recognition model differed in different *Tetrahymena* species, and how and why this recognition mechanism evolved and how its evolution contributed to speciation.

MTRC is likely to be a giant protein complex which is over a million Daltons in size. The large protein complex formed is reminiscent of the fusion patches that develop in budding or fission yeasts. In these species, the mating-type receptors are activated by ligand pheromones from the opposite mating type that induce polarity patch formation (see *Sieber et al., 2023*, for a recent review). At these patches, growth (shmooing) and fusion occur, which is reminiscent (in a different order) of the tip transformation in *T. thermophila*. Future studies of this protein complex may reveal whether more similar processes and/or mechanisms shared in these two distant lineages.

We still do not know what intracellular signals are transduced when the MTRCs on two cells interact. Mating-type self-recognition might generate an inhibitory signal or might simply inactivate the MTRC. In many species, such as *Papaver rhoeas* and *Ciona intestinalis*, interaction between mating-type proteins of the same mating type induces changes in cytoplasmic $Ca^{2+}$ concentration that cause self-incompatibility (*Giamarchi et al., 2006*; *Harada et al., 2008*; *Wu et al., 2011*). A similar system may exist in *T. thermophila*, since the MRC4 and MRC5 proteins are predicted to be $Ca^{2+}$-translocators. Interaction between MTRCs on cells of different mating types (non-self-recognition) should result in their activation to allow the cells to initiate pairing. We propose that the activation signal involves the CDK19 complex (a cyclin-dependent kinase complex) (*Ma et al., 2020*) and AKM3 (a $K^+$ channel; *Figure 3C*) because they probably interact with the MTRC. We expect future studies to lead to the discovery of more detailed mechanisms for mating-type recognition and initiation of conjugation involving these proteins. An intriguing discovery is that the *MTB2-eGFP* cell (VI background) can undergo selfing and mate with cells of all other mating types except VI and II. The inability to mate with VI and II supports the self-recognition model. Additionally, in theory, there should be two types of MTRCs in the *MTB2-eGFP* cell: MTRC of pure VI specificity (A6B6) and heterotypic MTRC of MTA6 and MTB2 (A6B2). Consequently, opposite mating-type recognition outcomes should occur simultaneously when the *MTB2-eGFP* cell recognizes another *MTB2-eGFP* cell: inhibition (A6B6–A6B6) and activation (A6B2–A6B2). The overall observed result showed that this strain can self, indicating there is no (MTRC is only blocked) or relatively weak inhibitory signal generated during self-recognition.

*Paramecium tetraurelia*, a closely related Oligohymenophorean ciliate, has only two mating types, which are determined by the expression or non-expression of a *Tetrahymena*-MTA/B-like protein called mtA (*Singh et al., 2014*; *Yan et al., 2021*). An intriguing question is whether mtA also engages in the formation of a MTRC with other proteins by serving as a recognizer rather than mediating a straightforward receptor–ligand interaction. Future investigations of *P. tetraurelia* may shed light on the origins and evolutionary aspects of this distinctive mating system. Moreover, due to the extremely long evolutionary distance, recognition mechanisms discovered in model species fall short of explaining many of the intricate biological events in protists. Insight into the detailed function of

MTRC could contribute to our understanding of cell–cell recognition processes in other species, such as *Toxoplasma* and *Plasmodium*.

# Materials and methods
## Biological methods

Strains used in this study are summarized in *Supplementary file 1*. All cell growth, starvation, and pairing experiments were conducted at 30°C. Cells were grown in Super Proteose Peptone (SPP) medium (1% Proteose Peptone, 0.1% yeast extract, 0.2% glucose, 0.003% Sequestrene) or Neff medium (0.25% Proteose Peptone, 0.25% yeast extract, 0.5% glucose, 0.003% Sequestrene). Cells were starved in 10 mM Tris-Cl (pH 7.4) for ~16 hr before all pairing experiments. For normal pairing assays, equal numbers of starved cells of different mating types (at ~2 × 10⁶ cells/ml) were mixed. To obtain costimulated (pre-incubated) cells, two starved strains were mixed at a 9:1 ratio for ~30 min (unless otherwise stated). Before mixing costimulated cells, any potentially pairing cells were separated by shaking. *Figure 2A* shows the setup of costimulation experiments. For all mating experiments (whether or not they involved mutant cells), the starting WT cell density was ~2 × 10⁶ cells/ml. To prepare starvation medium containing mating-essential factors, *Tetrahymena* cells (~2 × 10⁶ cells/ml) were starved in fresh medium for ~16 hr. Subsequently, cells were removed through three rounds of centrifugation (1000 × *g*, 3 min each). The following formula was used to calculate pairing ratios and correct for the presence of mutant cells:

$$\% \text{cells paired} = \frac{2 \times \# \, \text{pairs}}{\left(2 \times \# \, \text{pairs} + \# \, \text{unpaired cells}\right) \, \times \, \%\text{WT cells}} \times 100$$

## Somatic gene deletion, truncation, and protein tagging

To construct deletion strains, an ~1 kb fragment upstream of the gene's open reading frame (ORF) (#1), an ~0.5 kb fragment downstream of the gene's ORF (#2), and an ~1 kb fragment downstream of #2 (#3) were amplified. Fragments #2 and #3 were joined to the *NEO4* cassette (Cd²⁺-inducible *MTT1* promoter linked to the neomycin resistance gene) by fusion PCR and then cloned into the pBlueScript SK (+) vector together with fragment #1. In this way, #1-#2-*NEO4*-#3 constructs were obtained for the next transformation. HA-tagged strains were constructed in a similar way, except that fragment #1 was upstream of the stop codon or upstream of the terminal intron. To obtain the MTB2-eGFP construct, MTB2-coding sequences replaced the MTT1-coding sequence (*Figure 4E*) and the construct was made using the large DNA fragment assembly method (*Jiang et al., 2022*). Constructs were introduced into starved WT cells by biolistic transformation to obtain deletion strains (*Mochizuki, 2008*). Positive clones were selected in SPP medium containing decreasing Cd²⁺ concentrations (from 1 µg/ml to 0.05 µg/ml) and increasing paromomycin concentrations (from 120 µg/ml to 40 mg/ml) until all WT somatic chromosomes had been replaced by mutant ones, as determined by PCR using checking primers. All primers used are listed in *Supplementary file 2*.

## Immunoprecipitation and mass spectrometry

The IP method was adapted from a published method (*Tian et al., 2017*). To pull down HA-tagged proteins from *T. thermophila*, cells were harvested from 500 ml cultures (density ~3 × 10⁶ cells/ml). Cells were then treated for 20 min with paraformaldehyde (PFA) (at a final concentration of 0.3%) to stabilize protein–protein interactions, washed with PB buffer (2.7 mM KCl, 8 mM Na₂HPO₄, 1.5 mM K₂HPO₄), and blocked with 125 mM glycine. Cells were then resuspended in lysis buffer (1% Triton X-100, 30 mM Tris-HCl, 20 mM KCl, 2 mM MgCl₂, 1 mM phenylmethylsulfonyl fluoride, 150 mM NaCl, cOmplete proteinase inhibitor [Roche Diagnostics, Indianapolis, IN, USA]), lysed by ultrasonic treatment and incubated with EZview anti-HA agarose beads (Sigma-Aldrich, St Louis, MO, USA) for 2.5 hr at 4°C. The beads were washed with wash buffer (1% Triton X-100, 600 mM NaCl, 30 mM Tris-HCl, 20 mM KCl, 2 mM MgCl₂, cOmplete proteinase inhibitor) to remove nonspecific-binding proteins and then HA-tagged proteins were eluted with HA peptides (Sigma-Aldrich). WT samples (not HA-tagged) were run in parallel for each sample. In total, data for 13 WT controls were combined to identify nonspecific binding proteins.

For MS, the EASY-nLC chromatography system (Thermo Scientific, Rockford, IL, USA) was coupled online to an Orbitrap Elite instrument (Thermo Scientific) via a Nanospray Flex Ion Source (Thermo Scientific). Xtract software (Thermo Scientific) and Proteome Discoverer 2.1 software were used for MS data analysis based on a database that combines the 2014 version of whole genome protein annotation (http://ciliate.org/index.php/home/downloads, which contains the whole length sequence of MTA6 and MTB6) and mating-type-specific regions of all other mating-type proteins. IP data were analyzed using CRAPome (*Mellacheruvu et al., 2013*).

### Membrane protein extraction

*Figure 4—figure supplement 1A* shows the workflow used for membrane protein extraction. Cells were collected, resuspended in 20 ml lysis buffer (150 mM NaCl, 25 mM HEPES, 10% glycerol, 2 mM PMSF, 2.6 µg/ml aprotinin, 1.4 µg/ml pepstatin, 10 µg/ml leupeptin, pH 7.4), and lysed by high-pressure homogenization. The lysate was clarified first at low speed (14,000 rpm, 4°C, 15 min), and then at high speed (150,000 × $g$, 4°C, 1 hr). The pellet was resuspended in 5 ml lysis buffer containing 1% DDM (Anatrace, Maumee, OH, USA) and rotated for 2 hr at 4°C to extract the membrane proteins. Undissolved material was removed by centrifugation (14,000 rpm, 4°C, 30 min). The membrane protein extract was incubated with EZview anti-HA agarose beads for 2.5 hr at 4°C and then washed with 5 ml lysis buffer.

### Biotinylation and isolation of cell surface proteins

Pierce Cell Surface Protein Biotinylation and Isolation Kit (Thermo Scientific) was used to biotinylate and isolate cell surface proteins. For this, 75 ml cells (density ~3 × $10^6$ cells/ml) were harvested and washed once with BupH phosphate-buffered saline (PBS; 137 mM NaCl, 2.7 mM KCl, 4.3 mM $Na_2HPO_4$, 1.4 mM $KH_2PO_4$). Cells were then resuspended in 75 ml PBS containing 0.72 mg/ml Sulfo-NHS-SS-biotin and incubated at room temperature for 10 min. After two washes with 50 ml ice-cold BupH Tris buffer, cells were resuspended in 3 ml lysis buffer (PBS containing 1% Triton X-100, 1 mM phenylmethylsulfonyl fluoride, and cOmplete proteinase inhibitor [Roche Diagnostics]), lysed by ultrasonic treatment and incubated with 1.2 ml NeutrAvidin Agarose for 0.5 hr at room temperature. The resin was washed four times with 0.5 ml wash buffer and then cell surface proteins were eluted with 1.2 ml elution buffer (with 10 mM DTT). Before WB, cell surface protein samples were concentrated into 0.1 ml volumes using a 30 kDa centrifugal concentrator (Merck Millipore).

### Ciliary protein collection

To remove cilia, 500 ml cells (density ~3 × $10^6$ cells/ml) were harvested at room temperature and resuspended in 25 ml 10 mM Tris-Cl (pH 7.4), to which 50 ml medium A (10 mM $EDTA_2Na$, 50 mM sodium acetate, pH 6.0) was added. After 30 s, 25 ml cold distilled water was added; 1 min later, 0.25 ml 0.4 M $CaCl_2$ was added and incubated for 15 s. The cilia were detached from the calcium-shocked cells by vortexing three times for 5 s at 15 s intervals. To collect the cilia, cell bodies were removed by two rounds of centrifugation at 1500 rpm for 5 min at 4°C, and then cilia were collected by centrifugation at 15,000 rpm for 15 min at 4°C.

### Cytological methods

Fluorescein-labeled Con-A labeling was performed as previously reported (*Ma et al., 2020*). In brief, cells were fixed and stained with fluorescein-labeled Con-A (Vector Laboratories, Burlingame, CA, USA) at 37.5 µg/ml for 5 min and then washed three times with PB buffer.

For analysis of Tip transformation, cells were observed and photographed as soon as possible after fixation with 1% PFA. To distinguish between cell strains in a pairing mixture, starved cells of one strain were labeled with 500 nM MitoTracker (Invitrogen, Eugene, OR, USA), followed by two washes with 10 mM Tris-Cl (pH 7.4) before mixing.

For tubulin staining, cells were collected and fixed in PHEM buffer (30 mM PIPES, 14 mM HEPES, 5 mM EGTA, and 2 mM $MgSO_4$) containing 1% PFA and incubated for 30 min at 4°C. After three washes with PBS (10 min each), Tubulin-Atto 594 was added and incubated for 1 hr at 25°C. Finally, cells were washed three times with PBS (10 min each).

### Expression and purification of the extracellular region of mating-type proteins

Coding sequences of the extracellular region of mating-type proteins (MTA6xc, MTB6xc, MTA7xc, and MTB7xc) were codon-optimized and synthesized for expression in an insect cell system (*Trichoplusia*

*ni* Hi5 cells). Codon-optimized sequences were cloned into pFastBac vectors containing an N-terminal hemolin signal peptide sequence and a C-terminal 10× His tag sequence. The obtained constructs were transformed into competent DH10Bac cells and individual bacmids were transfected into *Spodoptera frugiperda* Sf9 cells. Recombinant baculoviruses were collected after 4 days and used to infect *T. ni* Hi5 cells for protein expression. Proteins were harvested 60 hr after infection and purified with Ni-NTA Superflow resin (QIAGEN), anion-exchange chromatography (Source 15Q, GE Healthcare), and size-exclusion chromatography (Superdex-200 Increase 10/300, GE Healthcare).

## Bioinformatics analysis

All microarray data were derived from *Tetra*FGD (*Xiong et al., 2011*; http://tfgd.ihb.ac.cn/). DNA sequencing data for mating type II–VII cells are derived from a previous report (*Wang et al., 2020*). Compute pI/Mw (https://web.expasy.org/compute_pi/) was used to predict protein molecular weight. InterProScan (http://www.ebi.ac.uk/interpro/) was used for function and domain annotation (*Jones et al., 2014*).

## Statistical analysis

For mating experiments, more than 100 unpaired cells or cell pairs were counted, with three to five independent replicates. GraphPad software (version 8.0.2) was used for statistical analysis based on ANOVA (matched, Fisher's LSD test).

## Acknowledgements

We thank Eduardo Orias (University of California Santa Barbara), Eileen P Hamilton (University of California Santa Barbara), and Kazufumi Mochizuki (University of Montpellier) for their help and suggestions about experimental design and manuscript writing, and Yunfei Wei (Huazhong University of Science and Technology) for tubulin staining using Tubulin-Atto 594. We also thank members of the Protist 10,000 Genomes Project (P10K) consortium for helpful suggestions. We would like to thank Min Wang at the Analysis and Testing Center of Institute of Hydrobiology, Chinese Academy of Sciences, for her help with mass spectrometry. The bioinformatics analysis was supported by the Wuhan Branch, Supercomputing Center, Chinese Academy of Sciences, China. Culture and maintenance of *Tetrahymena* cells were supported by the National Aquatic Biological Resource Center (NABRC).

---

## Additional information

### Funding

| Funder | Grant reference number | Author |
|---|---|---|
| National Natural Science Foundation of China | 32130011 | Wei Miao |
| Bureau of Frontier Sciences and Education, Chinese Academy of Sciences | ZDBS-LY-SM026 | Wei Miao |
| National Natural Science Foundation of China | 32200344 | Guanxiong Yan |
| China Postdoctoral Science Foundation | 2021M703433 | Guanxiong Yan |

The funders had no role in study design, data collection and interpretation, or the decision to submit the work for publication.

### Author contributions

Guanxiong Yan, Conceptualization, Data curation, Formal analysis, Funding acquisition, Validation, Investigation, Visualization, Writing - original draft, Writing – review and editing; Yang Ma, Investigation, Visualization, Writing – review and editing; Yanfang Wang, Jing Zhang, Haoming Cheng, Fanjie Tan, Su Wang, Investigation; Delin Zhang, Ping Yin, Supervision, Writing – review and editing; Jie

Xiong, Writing – review and editing; Wei Miao, Conceptualization, Supervision, Writing – review and editing

### Author ORCIDs
Wei Miao http://orcid.org/0000-0003-3440-8322

Reviewer #1 (Public Review): https://doi.org/10.7554/eLife.93770.3.sa1
Reviewer #2 (Public Review): https://doi.org/10.7554/eLife.93770.3.sa2
Reviewer #3 (Public Review): https://doi.org/10.7554/eLife.93770.3.sa3
Author Response https://doi.org/10.7554/eLife.93770.3.sa4

---

## Additional files

### Supplementary files
- Supplementary file 1. Strains used in this study.
- Supplementary file 2. Primers used in this study.
- MDAR checklist

### Data availability
All data generated or analysed during this study are included in the manuscript and supporting files.

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
