## [Editor Report · eLife assessment]

This **fundamental** study provides insight into the fascinating process of self- and non-self-recognition in the protist *Tetrahymena thermophila*, a species with seven distinct mating types. Using an elegant combination of phenotypic assays, protein studies, and imaging, the authors present **convincing** evidence that a large multifunctional protein complex at the cell surface mediates both self- and non-self mating-type recognition. This study extends our understanding of how more than two mating types/sexes may be specified in a species, and it will be relevant for anyone interested in sexual systems and cell-cell communication.

---

## [Referee Report · Reviewer #1 (Public Review)]

Summary:

In this study, Yan et al. investigate the molecular bases underlying mating type recognition in *Tetrahymena thermophila*. This model protist possesses a total of 7 mating types/sexes and mating occurs only between individuals expressing different mating types. The authors aimed to characterize the function of mating type proteins (MTA and MTB) in the process of self- and non-self recognition, using a combination of elegant phenotypic assays, protein studies, and imaging. They showed that the presence of MTA and MTB in the same cell is required for the expression of concavalin-A receptors and for tip transformation - two processes that are characteristic of the costimulation phase that precedes cell fusion. Using protein studies, the authors identify a set of additional proteins of varied functions that interact with MTA and MTB and are likely responsible for the downstream signaling processes required for mating. This is a description of a fascinating self- and non-self-recognition system and, as the authors point out, it is a rare example of a system with numerous mating types/sexes. This work opens the door for the further understanding of the molecular bases and evolution of these complex recognition systems within and outside protists.

The results shown in this study point to the unequivocal requirement of MTA and MTB proteins for mating. Nevertheless, some of the conclusions regarding the mode of functioning of these proteins are not fully supported and require additional investigation.

Strengths:

(1) The authors have established a set of very useful knock-out and reporter lines for MT proteins and extensively used them in sophisticated and well-designed phenotypic assays that allowed them to test the role of these proteins in vivo.

(2) Despite their apparent low abundance, the authors took advantage of a varied set of protein isolation and characterization techniques to pinpoint the localization of MT proteins to the cell membrane, and their interaction with multiple other proteins that could be downstream effectors. This opens the door for the future characterization of these proteins and further elucidation of the mating type recognition cascade.

Weaknesses:

The manuscript is structured and written in a very clear and easy-to-follow manner. However, several conclusions and discussion points fall short of highlighting possible models and mechanisms through which MT proteins control mating type recognition:

(1) The authors dismiss the possibility of a "simple receptor-ligand system", even though the data does not exclude this possibility. The model presented in Figure 2 S1, and on which the authors based their hypothesis, assumes the independence of MTA and MTB proteins in the generation of the intracellular cascade. However, the results presented in Figure 2 show that both proteins are required to be active in the same cell. Coupled with the fact that MTA and MTB proteins interact, this is compatible with a model where MTA would be a ligand and MTB a receptor (or vice-versa), and could thus form a receptor-ligand complex that could potentially be activated by a non-cognate MTA-MTB receptor-ligand complex, leading to an intracellular cascade mediated by the identified MRC proteins. As it stands, it is not clear what is the proposed working model, and it would be very beneficial for the reader for this to be clarified by having the point of view of the authors on this or other types of models.

(2) The presence of MTA/MTB proteins is required for costimulation (Figure 2), and supplementation with non-cognate extracellular fragments of these proteins (MTAxc, or MTBxc) is a positive stimulator of pairing. However, alone, these fragments do not have the ability to induce costimulation (Figure 5). Based on the results in Figures 5 and 6 the authors suggest that MT proteins mediate both self and non-self recognition. Why do MTAxc and MTBxc not induce costimulation alone? Are any other components required? How to reconcile this with the results of Figure 2? A more in-depth interpretation of these results would be very helpful, since these questions remain unanswered, making it difficult for the reader to extract a clear hypothesis on how MT proteins mediate self- and non-self-recognition.

---

## [Referee Report · Reviewer #2 (Public Review)]

This manuscript reports the discovery and analysis of a large protein complex that controls mating type and sexual reproduction of the model ciliate *Tetrahymena thermophila*. In contrast to many organisms that have two mating types or two sexes, Tetrahymena is multi-sexual with 7 distinct mating types. Previous studies identified the mating type locus, which encodes two transmembrane proteins called MTA and MTB that determine the specificity of mating type interactions. In this study, mutants are generated in the MTA and MTB genes and mutant isolates are studied for mating properties. Cells missing either MTA or MTB failed to co-stimulate wild-type cells of different mating types. Moreover, a mixture of mutants lacking MTA or MTB also failed to stimulate. These observations support the conclusion that MTA and MTB may form a complex that directs mating-type identity. To address this, the proteins were epitope-tagged and subjected to IP-MS analysis. This revealed that MTA and MTB are in a physical complex, and also revealed a series of 6 other proteins (MRC1-6) that together with MTA/B form the mating type recognition complex (MTRC). All 8 proteins feature predicted transmembrane domains, three feature GFR domains, and two are predicted to function as calcium transporters. The authors went on to demonstrate that components of the MTRC are localized on the cell surface but not in the cilia. They also presented findings that support the conclusion that the mating type-specific region of the MTA and MTB genes can influence both self- and non-self-recognition in mating.

Taken together, the findings presented are interesting and extend our understanding of how organisms with more than two mating types/sexes may be specified. The identification of the six-protein MRC complex is quite intriguing. It would seem important that the function of at least one of these subunits be analyzed by gene deletion and phenotyping, similar to the findings presented here for the MTA and MTB mutants. A straightforward prediction might be that a deletion of any subunit of the MRC complex would result in a sterile phenotype. The manuscript was very well written and a pleasure to read.

---

## [Referee Report · Reviewer #3 (Public Review)]

The authors describe the role, location, and function of the MTA and MTB mating type genes in the multi-mating-type species *T. thermophila*. The ciliate is an important group of organisms to study the evolution of mating types, as it is one of the few groups in which more than two mating types evolved independently. In the study, the authors use deletion strains of the species to show that both mating types genes located in each allele are required in both mating individuals for successful matings to occur. They show that the proteins are localized in the cell membrane, not the cilia, and that they interact in a complex (MTRC) with a set of 6 associated (non-mating type-allelic) genes. This complex is furthermore likely to interact with a cyclin-dependent kinase complex. It is intriguing that T. thermophila has two genes that are allelic and that are both required for successful mating. This coevolved double recognition has to my knowledge not been described for any other mating-type recognition system. I am not familiar with experimental research on ciliates, but as far as I can judge, the experiments appear well performed and mostly support the interpretation of the authors with appropriate controls and statistical analyses.

The results show clearly that the mating type genes regulate non-self-recognition, however, I am not convinced that self-recognition occurs leading to the suppression of mating. An alternative explanation could be that the MTA and MTB proteins form a complex and that the two extracellular regions together interact with the MTA+MTB proteins from different mating types. This alternative hypothesis fits with the coevolution of MTA and MTB genes observed in the phylogenetic subgroups as described by Yan et al. (2021 iScience). Adding MTAxc and/or MTBxc to the cells can lead to the occupation of the external parts of the full proteins thereby inhibiting the formation of the complex, which in turn reduces non-self interactions. Self-recognition as explained in Figure 2S1 suggests an active response, which should be measurable in expression data for example. This is in my opinion not essential, but a claim of self-recognition through the MTA and MTB should not be made.

The authors discuss that *T. thermophila* has special mating-type proteins that are large, while those of other groups are generally small (lines 157-160 and discussion). The complex formed is very large and in the discussion, they argue that this might be due to the "highly complex process, given that there are seven mating types in all". There is no argument given why large is more complex, if this is complex, and whether more mating types require more complexity. In basidiomycete fungi, many more mating types than 7 exist, and the homeodomain genes involved in mating types are relatively small but highly diverse (Luo et al. 1994 PMID: 7914671). The mating types associated with GPCR receptors in fungi are arguably larger, but again their function is not that complex, and mating-type specific variations appear to evolve easily (Fowler et al 2004 PMID: 14643262; Seike et al. 2015 PMID: 25831518). The large protein complex formed is reminiscent of the fusion patches that develop in budding or fission yeasts. In these species, the mating type receptors are activated by ligand pheromones from the opposite mating type that induce polarity patch formation (see Sieber et al. 2023 PMID: 35148940 for a recent review). At these patches, growth (shmooing) and fusion occur, which is reminiscent (in a different order) of the tip transformation in T. thermophilia. The fusion of two cells is in all taxa a dangerous and complex event that requires the evolution of very strict regulation and the existence of a system like the MTRC and cyclin-dependent complex to regulate this process is therefore not unexpected. The existence of multiple mating types should not greatly complicate the process, as most of the machinery (except for the MTA and MTB) is identical among all mating types.

The Tetrahymena/ciliate genetics and lifecycle could be better explained. For a general audience, the system is not easy to follow. For example, the ploidy of the somatic nucleus with regards to the mating type is not clear to me. The MAC is generally considered "polyploid", but how does this work for the mating type? I assume only a single copy of the mating type locus is available in the MAC to avoid self-recognition in the cells. Is it known how the diploid origin reduces to a single mating type? This does not become apparent from Cervantes et al. 2013. Also, the explanation of co-stimulation is not completely clear (lines 49-60). Initially, direct cell-cell contact is mentioned, but later it is mentioned that "all cells become fully stimulated", even when unequal ratios are used. Is physical contact necessary? Or is this due to the "secrete mating-essential factors" (line 601)? These details are essential, for interpretation of the results and need to be explained better.

Abstract and introduction: Sexes are not mating types. In general, mating types refer to systems in which there is no obvious asymmetry between the gametes, beyond the compatibility system. When there is a physiological difference such as size or motility, sexes are used. This distinction is of importance because in many species mating types and sexes can occur together, with each sex being able to have either (when two) or multiple mating types. An example are SI in angiosperms as used as an example by the authors or mating types in filamentous fungi. See Billiard et al. 2011 [PMID: 21489122] for a good explanation and argumentation for the importance of making this distinction.

---

## [Author Response]

The following is the authors’ response to the original reviews.

**Public Reviews:**

**Reviewer #1:**
Summary:In this study, Yan et al. investigate the molecular bases underlying mating type recognition in *Tetrahymena thermophila*. This model protist possesses a total of 7 mating types/sexes and mating occurs only between individuals expressing different mating types. The authors aimed to characterize the function of mating type proteins (MTA and MTB) in the process of self- and non-self recognition, using a combination of elegant phenotypic assays, protein studies, and imaging. They showed that the presence of MTA and MTB in the same cell is required for the expression of concavalin-A receptors and for tip transformation - two processes that are characteristic of the costimulation phase that precedes cell fusion. Using protein studies, the authors identify a set of additional proteins of varied functions that interact with MTA and MTB and are likely responsible for the downstream signaling processes required for mating. This is a description of a fascinating self- and non-self-recognition system and, as the authors point out, it is a rare example of a system with numerous mating types/sexes. This work opens the door for the further understanding of the molecular bases and evolution of these complex recognition systems within and outside protists.The results shown in this study point to the unequivocal requirement of MTA and MTB proteins for mating. Nevertheless, some of the conclusions regarding the mode of functioning of these proteins are not fully supported and require additional investigation.Strengths:(1) The authors have established a set of very useful knock-out and reporter lines for MT proteins and extensively used them in sophisticated and well-designed phenotypic assays that allowed them to test the role of these proteins in vivo.(2) Despite their apparent low abundance, the authors took advantage of a varied set of protein isolation and characterization techniques to pinpoint the localization of MT proteins to the cell membrane, and their interaction with multiple other proteins that could be downstream effectors. This opens the door for the future characterization of these proteins and further elucidation of the mating type recognition cascade.Weaknesses:The manuscript is structured and written in a very clear and easy-to-follow manner. However, several conclusions and discussion points fall short of highlighting possible models and mechanisms through which MT proteins control mating type recognition:(1) The authors dismiss the possibility of a "simple receptor-ligand system", even though the data does not exclude this possibility. The model presented in Figure 2 S1, and on which the authors based their hypothesis, assumes the independence of MTA and MTB proteins in the generation of the intracellular cascade. However, the results presented in Figure 2 show that both proteins are required to be active in the same cell. Coupled with the fact that MTA and MTB proteins interact, this is compatible with a model where MTA would be a ligand and MTB a receptor (or vice-versa), and could thus form a receptor-ligand complex that could potentially be activated by a non-cognate MTA-MTB receptor-ligand complex, leading to an intracellular cascade mediated by the identified MRC proteins. As it stands, it is not clear what is the proposed working model, and it would be very beneficial for the reader for this to be clarified by having the point of view of the authors on this or other types of models.

We are very grateful that Reviewer #1 proposed the possibility that MTA and MTB form a receptor-ligand complex in which one acting as the ligand and the other as the receptor. We considered this hypothesis when asking how dose MTRC function, too. However, our current results do not support this idea. For instance, if MTA were a ligand and MTB a receptor, we would expect a mating signal upon treatment with MTAxc protein, but not with MTBxc. Contrary to this expectation, our experiments revealed that both MTAxc and MTBxc exhibit very similar effects (Figure 5, green and blue), and their combined treatment produces a stronger effect (Figure 5, teal). This suggests a mixed function for both proteins. (We incorporated this discussion into the revised version [line 120-121, 240-244].) It is pity that our current knowledge does not provide a detailed molecular mechanism for this intricate system. We are actively investigating the protein structures of MTA, MTB, and the entire MTRC, hoping to gain deeper insights into the molecular functions of MTA and MTB.

Additionally, we also realized that the expression we used in the previous version, “simple receptor-ligand model”, is not clearly defined. As Reviewer #1 pointed out, in this section, we examined whether the individual proteins of MTA and MTB act as a couple of receptor and ligand. We think this is the simplest possibility as a null hypothesis for Tetrahymena mating-type recognition. We have clarified it in the revised version (line 90-91, 104-106). According to this section, we proposed that MTA and MTB may form a complex that serves as a recognizer (functioning as both ligand and receptor) (line 117-118).

(2) The presence of MTA/MTB proteins is required for costimulation (Figure 2), and supplementation with non-cognate extracellular fragments of these proteins (MTAxc, or MTBxc) is a positive stimulator of pairing. However, alone, these fragments do not have the ability to induce costimulation (Figure 5). Based on the results in Figures 5 and 6 the authors suggest that MT proteins mediate both self and non-self recognition. Why do MTAxc and MTBxc not induce costimulation alone? Are any other components required? How to reconcile this with the results of Figure 2? A more in-depth interpretation of these results would be very helpful, since these questions remain unanswered, making it difficult for the reader to extract a clear hypothesis on how MT proteins mediate self- and non-self-recognition.

Several factors could contribute to the inability of MTA/Bxc to induce costimulation. It is highly likely that additional components are necessary, given that MTA/B form a protein complex with other proteins. Moreover, the expression of MTA/Bxc in insect cells, compared with Tetrahymena, might result in differences in post-translational modifications. Additionally, there are variations in protein conditions; on the Tetrahymena membrane, these proteins are arranged regularly and concentrated in a small area, while MTA/Bxc is randomly dispersed in the medium. The former condition could be more efficient. If there is a threshold required to stimulate a costimulation marker, MTA/Bxc may fail to meet this requirement. Much more studies are needed to fully answer this question. We acknowledged this limitation in the revised version (line 244-248).

**Reviewer #2:**
This manuscript reports the discovery and analysis of a large protein complex that controls mating type and sexual reproduction of the model ciliate *Tetrahymena thermophila*. In contrast to many organisms that have two mating types or two sexes, Tetrahymena is multi-sexual with 7 distinct mating types. Previous studies identified the mating type locus, which encodes two transmembrane proteins called MTA and MTB that determine the specificity of mating type interactions. In this study, mutants are generated in the MTA and MTB genes and mutant isolates are studied for mating properties. Cells missing either MTA or MTB failed to co-stimulate wild-type cells of different mating types. Moreover, a mixture of mutants lacking MTA or MTB also failed to stimulate. These observations support the conclusion that MTA and MTB may form a complex that directs mating-type identity. To address this, the proteins were epitope-tagged and subjected to IP-MS analysis. This revealed that MTA and MTB are in a physical complex, and also revealed a series of 6 other proteins (MRC1-6) that together with MTA/B form the mating type recognition complex (MTRC). All 8 proteins feature predicted transmembrane domains, three feature GFR domains, and two are predicted to function as calcium transporters. The authors went on to demonstrate that components of the MTRC are localized on the cell surface but not in the cilia. They also presented findings that support the conclusion that the mating type-specific region of the MTA and MTB genes can influence both self- and non-self-recognition in mating.Taken together, the findings presented are interesting and extend our understanding of how organisms with more than two mating types/sexes may be specified. The identification of the six-protein MRC complex is quite intriguing. It would seem important that the function of at least one of these subunits be analyzed by gene deletion and phenotyping, similar to the findings presented here for the MTA and MTB mutants. A straightforward prediction might be that a deletion of any subunit of the MRC complex would result in a sterile phenotype. The manuscript was very well written and a pleasure to read.

Thanks for the valuable comments and suggestions. We are currently in the process of constructing deletion strains for these genes. As of now, we have successfully obtained ΔMRC1-3 and MRC4-6 knockdown strains. Our preliminary observations indicate that ΔMRC1-3 strains are unable to undergo mating. However, we prefer not to include these results in the current manuscript, as we believe that more comprehensive studies are still needed.

**Reviewer #3:**
The authors describe the role, location, and function of the MTA and MTB mating type genes in the multi-mating-type species *T. thermophila*. The ciliate is an important group of organisms to study the evolution of mating types, as it is one of the few groups in which more than two mating types evolved independently. In the study, the authors use deletion strains of the species to show that both mating types genes located in each allele are required in both mating individuals for successful matings to occur. They show that the proteins are localized in the cell membrane, not the cilia, and that they interact in a complex (MTRC) with a set of 6 associated (non-mating type-allelic) genes. This complex is furthermore likely to interact with a cyclin-dependent kinase complex. It is intriguing that T. thermophila has two genes that are allelic and that are both required for successful mating. This coevolved double recognition has to my knowledge not been described for any other mating-type recognition system. I am not familiar with experimental research on ciliates, but as far as I can judge, the experiments appear well performed and mostly support the interpretation of the authors with appropriate controls and statistical analyses.The results show clearly that the mating type genes regulate non-self-recognition, however, I am not convinced that self-recognition occurs leading to the suppression of mating. An alternative explanation could be that the MTA and MTB proteins form a complex and that the two extracellular regions together interact with the MTA+MTB proteins from different mating types. This alternative hypothesis fits with the coevolution of MTA and MTB genes observed in the phylogenetic subgroups as described by Yan et al. (2021 iScience). Adding MTAxc and/or MTBxc to the cells can lead to the occupation of the external parts of the full proteins thereby inhibiting the formation of the complex, which in turn reduces non-self interactions. Self-recognition as explained in Figure 2S1 suggests an active response, which should be measurable in expression data for example. This is in my opinion not essential, but a claim of self-recognition through the MTA and MTB should not be made.

We express our gratitude to Reviewer #3 for proposing the occupation model and have incorporated this possibility into the manuscript. We believe it is possible that occupation may serve as the molecular mechanism through which self-recognition negatively regulates mating. If there is a physical interaction between mating-type proteins of the same type, but this interaction blocks the recognition machinery rather than initiating mating, it can be considered a form of self-recognition. This aligns with the observation that strains expressing MTA/B6 and MTB2 mate normally with WT cells of all mating types except for VI and II (line 203-204). A concise discussion on this topic is included in the manuscript (line 288-293, 659-661). We are actively investigating the downstream aspects of mating-type recognition, and we hope to provide further insights into this question soon.

The authors discuss that *T. thermophila* has special mating-type proteins that are large, while those of other groups are generally small (lines 157-160 and discussion). The complex formed is very large and in the discussion, they argue that this might be due to the "highly complex process, given that there are seven mating types in all". There is no argument given why large is more complex, if this is complex, and whether more mating types require more complexity. In basidiomycete fungi, many more mating types than 7 exist, and the homeodomain genes involved in mating types are relatively small but highly diverse (Luo et al. 1994 PMID: 7914671). The mating types associated with GPCR receptors in fungi are arguably larger, but again their function is not that complex, and mating-type specific variations appear to evolve easily (Fowler et al 2004 PMID: 14643262; Seike et al. 2015 PMID: 25831518). The large protein complex formed is reminiscent of the fusion patches that develop in budding or fission yeasts. In these species, the mating type receptors are activated by ligand pheromones from the opposite mating type that induce polarity patch formation (see Sieber et al. 2023 PMID: 35148940 for a recent review). At these patches, growth (shmooing) and fusion occur, which is reminiscent (in a different order) of the tip transformation in T. thermophilia. The fusion of two cells is in all taxa a dangerous and complex event that requires the evolution of very strict regulation and the existence of a system like the MTRC and cyclin-dependent complex to regulate this process is therefore not unexpected. The existence of multiple mating types should not greatly complicate the process, as most of the machinery (except for the MTA and MTB) is identical among all mating types.

We are very grateful that Reviewer #3 provide this insightful view and relevant papers. In response to the feedback, we removed the sentences regarding “multiple mating types greatly complicate the process” in the revised version. Instead, we have introduced a discussion section comparing the mating systems of yeasts and Tetrahymena (line 279-286).

The Tetrahymena/ciliate genetics and lifecycle could be better explained. For a general audience, the system is not easy to follow. For example, the ploidy of the somatic nucleus with regards to the mating type is not clear to me. The MAC is generally considered "polyploid", but how does this work for the mating type? I assume only a single copy of the mating type locus is available in the MAC to avoid self-recognition in the cells. Is it known how the diploid origin reduces to a single mating type? This does not become apparent from Cervantes et al. 2013.

In *T. thermophila*, the MIC (diploid) contains several mating-type gene pairs (mtGP, i.e., MTA and MTB) organized in a tandem array at the mat locus on each chromosome. In sexual reproduction, the new MAC of the progeny develops from the fertilized MIC through a series of genome editing events, and its ploidy increases to ~90 by endoreduplication. During this process, mtGP loss occurs, resulting in only one mtGP remaining on the MAC chromosome. The mating-type specificity of mtGPs on each chromosome within one nucleus becomes relatively pure through intranuclear coordination. After multiple assortments (possibly caused by MAC amitosis during cell fission), only mtGPs of one mating-type specificity exist in each cell, determining the cell’s mating type.

It is pity that the exact mechanisms involved in this complicated process remain a black box. The loss of mating-type gene pairs is hypothesized to involve a series of homologous recombination events, but this has not been completely proven. Furthermore, there is no clear understanding of how intranuclear coordination and assortment are achieved. While we have made observations confirming these events, a breakthrough in understanding the molecular mechanism is yet to be achieved.

We included more information in the revised version (line 672-683). Given the complexity of these unusual processes, we recommend an excellent review by Prof. Eduardo Orias (PMID: 28715961), which offers detailed explanations of the process and related concepts (line 685-686).

Also, the explanation of co-stimulation is not completely clear (lines 49-60). Initially, direct cell-cell contact is mentioned, but later it is mentioned that "all cells become fully stimulated", even when unequal ratios are used. Is physical contact necessary? Or is this due to the "secrete mating-essential factors" (line 601)? These details are essential, for interpretation of the results and need to be explained better.

Sorry that we didn’t realize the term “contact” is not precise enough. In Tetrahymena, physical contact is indeed necessary, but it can refer to temporary interactions. Unlike yeast, Tetrahymena cells exhibit rapid movement, swimming randomly in the medium. Occasionally, two cells may come into contact, but they quickly separate instead of sticking together. Even newly formed loose pairs often become separated. As a result, one cell can come into contact with numerous others and stimulate them. We have clarified this aspect in the revised version (line 50-51, 57).

Abstract and introduction: Sexes are not mating types. In general, mating types refer to systems in which there is no obvious asymmetry between the gametes, beyond the compatibility system. When there is a physiological difference such as size or motility, sexes are used. This distinction is of importance because in many species mating types and sexes can occur together, with each sex being able to have either (when two) or multiple mating types. An example are SI in angiosperms as used as an example by the authors or mating types in filamentous fungi. See Billiard et al. 2011 [PMID: 21489122] for a good explanation and argumentation for the importance of making this distinction.

We have clarified the expression in the revised version (line 20, 38, 40, 45).

**Recommendations for the authors:**

**Reviewer #1:**
I really enjoyed reading this manuscript and I think a few tweaks in the writing/data presentation could greatly improve the experience for the reader:(1) The information about your previous work in identifying downstream proteins CDK19, CYC9, and CIP1 (lines 170-173) could be directly presented in the introduction.

We have moved this information in the introduction in the revised version (line 74-77).

(2) For a reader who is not familiar with Tetrahymena, a few more details on how reporter and knock-out lines are generated would be beneficial.

We introduced the knock-out method in Figure 2 – figure supplement 1B, HA-tag method in Figure 3A, and MTB2-eGFP construction method in Figure 4E. In addition, we introduced how co-stimulation markers observed in Materials and Methods (line 404-410)

(3) Figures 5 and 6: clarify the types of pairing and treatments that were done directly in the figure (eg. adding additional labels). As of now, it is necessary to go through the text and legend to try and understand in detail what was done.

Cell types and treatments were directly introduced in the revised figure (Figure 5 and 6).

(4) The logical transition in lines 136-142 is hard to follow.

We rewrote this paragraph in the revised version (lines 143-156). Additionally, we added a figure to illustrate the theoretical mating-type recognition model between WT cells and ΔCDK19, ΔCYC9 cells, MTAxc, MTBxc proteins, and ΔMTA, ΔMTB cells (Figure 2 – figure supplement 1D-G).

(5) Lines 191-196: the fact that cells expressing multiple mating types can self goes against an active self-rejection system - if this is the case there should be self-rejection among all expressed mating types. Unless non-self recognition is an active process and self-recognition is simply the absence of non-self recognition. The authors briefly mention this in lines 263-265, but it would be interesting to expand and clarify this.

We appreciate that Reviewer #1 notice the interesting selfing phenotype of the MTB2-eGFP (MTVI background) strain. We further discussed it in the revised manuscript (line 298-306).

(6) The authors briefly mention the possibility of different mating types using different recognition mechanisms (lines 255-260), based on the big differences in the size of the mating-specific region of MT proteins. Following this and the weakness nr. 2, I think it would be pertinent to gather and present more information on the properties and structures of the mating-type specific regions of MT proteins. Simple in silico analysis of motifs, structure, etc. could help clarify the role of these regions. It seems more parsimonious that MT proteins would have variable mating type specific regions that account for the recognition of the different mating types, and conserved cytoplasmic functions that could trigger a single downstream signaling cascade. It would be interesting to know the authors' opinion on this.

We are very grateful for this suggestion. Actually, we are currently working on determining the 3D structure of MTRC. The Alphafold2 prediction indicates that the MT-specific region is comprised of seven global β-sheets, resembling the structure of immunoglobulins (Ig). Our most recent cryo-EM results have revealed a ~15Å structure, aligning well with the prediction. However, the main challenge lies in the low expression levels, both in Tetrahymena and insect/mammal cells. We anticipate obtaining more detailed results soon. Therefore, we prefer to present the MT recognition model with robust experimental evidence in the future, and didn’t discuss too much on this aspect in the current manuscript.

(7) Adding a figure including a proposed model, as well as expanding the discussion on the points presented as "weaknesses" would help clarify the ideas/hypothesis on how the mating recognition works. I think this would really elevate the paper and help highlight the results.

We added a figure to introduce the model and the weaknesses in the revised version (Figure 7, line 656-665).

(8) Line 202-203: It is far-fetched to infer subcellular localization based on the data presented here, couterstaining with other dyes and antibodies specific to certain cell components, as well as negative control images, are required.

Thanks for the suggestion. We attempted to stain cell components using various dyes and antibodies. Unfortunately, we found that cell surface and cilia (especially oral cilia) is very easy to give a false positive signal. We think this issue seriously affects the credibility of the results. It may seem like splitting hairs, but we are trying to be precise.

Meanwhile, we still believe the mating-type proteins localizes to cell surface because MTA-HA is identified in the isolated cell surface proteins.

Regarding negative control, as shown in Fig. 4G, where a MTB2-eGFP cell is pairing with a WT cell, no GFP signal is observed in the WT cell.

(9) Lines 131: clarify the sentence - expression of Con-A receptors requires both MTA and MTB (MTA to receive the signal).

We modified the sentence in the revised version (line 139-140).

**Reviewer #2:**
Minor points.(1) Line 194-196. Why are these cells able to self?

These cells able to self may because the MTRC contain heterotypic mating-type proteins (MTA6 and MTB2), which activate mating when they interact with another heterotypic MTRC (line 207-208).

(2) Line 232. What do the authors mean by the term synergistic effect here? Definition and statistics?

Sorry about the confusion. The synergistic effect refers to the effect of MTAxc and MTBxc become stronger when using together. We clarified it in the revised version (line 232).

(3) For Figure 4 panel D, are there antibodies that are available as a control for cilia? If so, then blotting this membrane would show that cilia-associated proteins are in the cilia preparation, which is a standard control for sub-cellular fractionation.

Thanks for the suggestion. Unfortunately, we didn’t find a suitable cilia-specific antibody yet. Instead, we employed MS analysis to confirm the presence of cilia proteins in this sample (line 195-196, Figure 4–Source data 1). We also observed the sample under the microscope, which directly revealed the presence of cilia (Figure 4C).

(4) At least one reference cited in the text was not present in the reference list. The authors should go through the references cited to ensure that all have made it into the reference list.

We have checked all the references.

Some minor edits:

(1) MTA and MTB are presented in both roman and italics (e.g. line 209) in the manuscript. Maybe all should be in italics? Or is this a distinction between the gene and the protein?

The italics word (MTA) refers to gene, and non-italics word (MTA) refers to protein.

(2) Line 251. Change "achieving" to "achieve".

We have corrected this word (line 266).

**Reviewer #3:**
Line 101. It would help to explain this expectation earlier in this paragraph.

We explained the expectation in the revised version (line 92-97, 104-106).

Line 109. How is a co-receptor different from the MTRC complex?

We have rewritten the relevant sentences to enhance clarity (line 116-119). The molecular function of the MTRC complex could involve acting as a co-receptor or recognizer (functioning as both ligand and receptor). Based on the results presented in this section, we propose that MTA and MTB may function as a complex, but the confirmation of this hypothesis (MTRC) is provided in a later section. Therefore, we did not use the term “MTRC” here. These sentences briefly discuss the molecular function of this complex and explain why MTRC does not appear to function as a co-receptor.

Line 251: which "dual approach" is referred to?

Dual approach is referred to both self and non-self recognition. We explained it in the revised version (line 265-266).

Line 258: what "different mechanisms" do the authors have in mind? Why would a different mechanism be expected? The different sizes could have evolved for (coevolutionary?) selection on the same mechanism.

Sorry about the confusion. We clarified it in the revised version (line 269-278).

What we intended to express is that we are uncertain whether the mating-type recognition model we discovered in *T. thermophila* is applicable to all Tetrahymena species due to significant differences in the length of the mating-type-specific region. We believe it is important to highlight this distinction to avoid potential misinterpretations in future studies involving other Tetrahymena species. At the same time, we look forward to future research that may provide insights into this question.

Fig 2 C&D. Is it correct that these figures show the strains only after 'preincubation'? This is not apparent from the caption of the text. Additionally, the order of the images is very confusing. Write in the figures (so not just in the caption) what the sub-script means.

These panels are re-organized in the revised version (Fig. 2C&D). There are three kinds of pictures: “not incubated”, “WT pre-incubated by mutant” and “mutant pre-incubated by WT”.

The methods used to generate Figure 5 are not clearly described. I understand that the obtained xc proteins were added to the cells, and then washed, after which a test was performed mixing WT-VI and WT-VII cells. Were both cells treated? Or only one of the strains? The explanation for the reused washing medium is not clear and the method is not indicated.

Both cells are treated. More details are provided in the revised manuscript (line 230-231, 633-634, 637-639, Fig. 5). To prepare the starvation medium containing mating-essential factors, cells were starved in fresh starvation medium for ~16 hours. Subsequently, cells were removed by three rounds of centrifugation (1000 g, 3 min) (line 330-332).

In general, the figures are difficult to understand without repeated inquiries in the captions. Give more information in the figures themselves.

More information is introduced in the figure (Fig. 2C, Fig. 3B, Fig. 4A, B, D, Fig. 5 and Fig. 6).